# Pattern Pick and Place Method for Twisted Bi- and Multi-Layer Graphene

**DOI:** 10.3390/ma12223740

**Published:** 2019-11-13

**Authors:** Jae-Young Lim, Hyeon-Sik Jang, Hyun-Jae Yoo, Seung-il Kim, Dongmok Whang

**Affiliations:** 1SKKU Advanced Institute of Nanotechnology (SAINT) and School of Advanced Materials Science and Engineering Sungkyunkwan University (SSKU), 2066, Seobu-Ro, Jangan-Gu, Suwon-Si, Gyeonggi-Do 16419, Korea; limjyyy@skku.edu (J.-Y.L.); dagu1821@skku.edu (H.-S.J.); silverains2@gmail.com (H.-J.Y.); 2Department of Energy Systems Research and Department of Materials Science and Engineering Ajou University, 2016, World cup-Ro, Yeongtong-Gu, Suwon-Si, Gyeonggi-Do 16499, Korea; stmddlfs@gmail.com

**Keywords:** twisted bi-layer graphene, graphene transfer, 2D superlattice

## Abstract

Twisted bi-layer graphene (tBLG) has attracted much attention because of its unique band structure and properties. The properties of tBLG vary with small differences in the interlayer twist angle, but it is difficult to accurately adjust the interlayer twist angle of tBLG with the conventional fabrication method. In this study, we introduce a facile tBLG fabrication method that directly picks up a single-crystalline graphene layer from a growth substrate and places it on another graphene layer with a pre-designed twist angle. Using this approach, we stacked single-crystalline graphene layers with controlled twist angles and thus fabricated tBLG and twisted multi-layer graphene (tMLG). The structural, optical and electrical properties depending on the twist angle and number of layers, were investigated using transmission electron microscopy (TEM), micro–Raman spectroscopy, and gate-dependent sheet resistance measurements. The obtained results show that the pick and place approach enables the direct dry transfer of the top graphene layer on the as-grown graphene to fabricate uniform tBLG and tMLG with minimal interlayer contamination and pre-defined twist angles.

## 1. Introduction

Bi-layer graphene, in which two sheets of monolayer graphene are stacked, has unique physical and electrical properties that are different from monolayer graphene [1,2]. Typically, bi-layer graphene is a Bernal stack (AB stack) and has an adjustable bandgap with a perpendicular electric field [3]. Therefore, instead of single-layer graphene without a bandgap, AB stacked bi-layer graphene has been studied as a channel material of a new transistor structure [4]. Recently, twisted bilayer graphene, in which two sheets of monolayer graphene are stacked with a twist angle, has received considerable attention because its unique properties depend on a twist angle such as van Hove singularities (vHSs) [5], tunable optical conductivity [6], and renormalization of Fermi velocity [7,8]. In particular, the unconventional superconducting properties at magic angles [9], and quasi-crystalline structure at 30° [10] have proved that twisted bi-layer graphene (tBLG) is a promising material in various fields. Generally, tBLG has been fabricated by folding [6,11] or stacking [12,13,14] exfoliated graphene flakes. For example, polymer supporting layer has often been used for stacking graphene flakes and thus fabricating tBLG [13]. Chen et al. reported that a single crystal graphene flake, exfoliated from highly oriented pyrolytic graphite (HOPG), was cut into two pieces with a laser, and tBLG was fabricated by stacking the two pieces with a controlled twist angle [12]. The tBLG fabricated using graphene flakes has a limited size for its practical application and also, contains impurities, which are induced by direct contact with polymer [14]. Alternatively, bi-layer graphene directly synthesized by chemical vapor deposition (CVD) can increase in size, but the interlayer twist angle is not controllable, and it is difficult to produce uniform layers [5,15,16]. When tBLG is produced by stacking two single-crystal monolayers of graphene synthesized by CVD [17,18], accurate alignment of the interlayer twist angle is challenging, and the interlayer residues trapped during the transfer process hinder interlayer coupling [12]. Wang et al. used a hexagonal boron nitride (hBN) flake instead of polymeric materials as a supporting layer to transfer the graphene, which can reduce impurities becoming trapped between the layers [19]. Banszerus et al. reported a high-performance graphene device fabricated by the direct pick-up of the graphene layer grown on Cu substrate using the hBN-based van der Waals transfer [20]. However, this method only allows for the hBN-encapsulated heterostructures, and there is a limitation in size due to the use of hBN flakes. In addition, twisted multi-layer graphene (tMLG), which has more than two layers of graphene, may have significant potential for various functional devices but has rarely been studied [21,22,23].

In this study, we introduce a facile pick and place method for fabricating precise and uniform twist bi-layer and a multi-layer graphene structure with a clean interface. We were able to selectively separate the desired shapes of the graphene on as-grown large-area single-crystalline graphene by the pattern pick up process. After that, the tBLG was fabricated by rotating the angle and placing it on the as-grown graphene again. The van der Waals interaction of the tBLG fabricated on the catalyst substrate was stronger than the interaction between the catalyst substrate and the graphene, thus the tBLG could easily be separated from the catalyst substrate and transferred to the target substrate. The fabricated tBLG has an accurate and uniform twist angle, which was confirmed by Raman spectroscopy and transmission electron microscope analysis.

## 2. Materials and Methods 

### 2.1. Preparation of the Pattern on as-Grown Graphene

Graphene was synthesized on Germanium (Ge) (110) substrate by the CVD method [24]. E-beam lithography was used to draw a 40 × 40 μm^2^ square pattern on the graphene grown on Ge wafers. A 40 nm thick gold (Au) film was deposited onto the pattern using a thermal evaporator. Then, Au-patterned graphene substrate was annealed in a hydrogen (H_2_) and argon (Ar) atmosphere at 200 °C and 100 torr for 2 h.

### 2.2. Production of the Stamp 

A 10:1 ratio mixture of SYGARD 184 base and curing agent was dropped onto a slide glass and cured at 70 °C to produce a polydimethylsiloxane (PDMS) stamp. The surface of the PDMS stamp was hydrophilized by O_2_ plasma treatment (100 sccm, 100 A, 10 min) and polypropylene carbonate (PPC) was spin-coated (1500 R.P.M, 60 s) on the surface of the PDMS stamp. 

### 2.3. Pick and Place Process 

For the pick and place process, we used home-made transfer equipment consisting of an optical microscope, a sample stage (three-axis (X, Y, Z) manual stage with 360° rotation and hot chuck), and a stamp stage (three-axis manual stage). The PPC/PDMS stamp immobilized on the stamp stages was contacted with the gold pattern on the as-growTn graphene. Then, the temperature was slowly raised to 50 °C, and the stamp was lifted to exfoliate the gold pattern and the underlying graphene from the Ge substrate. The gold/graphene on the stamp was attached to the target substrate at 130 °C. As the adhesion between the PPC and gold/graphene weakened at the high temperature, gold/graphene could be placed on the target substrate. It was then soaked in acetone for 20 min to remove any polymer on the surface of the gold [25]. The remaining polymer residue on the gold surface was removed by O_2_ plasma cleaning (100 sccm, 100 A, 5 min). 

### 2.4. Transmission Electron Microscopy (TEM) Specimens Preparation

Polymethyl methacrylate (PMMA) was spin-coated on the tBLG/SiO_2_/Si 300 nm substrate at 2000 rpm for 1 min. The spin-coated substrate was baked at 100 °C for 5 min and floated on the etching solution (HF:H_2_O_2_:H_2_O = 1:1:1). As the SiO_2_ etched, the PMMA/tBLG was separated from the substrate and floated on the DI-water for cleaning. The PMMA/ tBLG was placed onto a TEM grid, and then the PMMA was removed by using acetone vapor and solution.

### 2.5. Device Fabrication

The back gated graphene field-effect transistor (FET) were fabricated on SiO_2_/Si 300 nm substrate by standard photolithography and metal electrode deposition (Cr 5 nm, Au 40 nm). The sheet resistance of the graphene FET was measured using a Keithley SCS-4200 system at room temperature. The charge carrier density was calculated by using a simple Drude model and the carrier mobilities were extracted at a high carrier concentration of *n* = 3 × 10^12^ cm^−2^.

### 2.6. Characterizations

The high-resolution transmission electron microscopy (HR–TEM) images and the selected area electron diffraction (SAED) patterns of graphene were taken using a JEOL ARM 200F (JEOL, LTD, Tokyo, Japan) at 80 kV. A Raman microscope (WITEC Raman system, Ulm, Germany) with excitation energy of 532 nm at the MEMS·Sensor Platform Center of Sungkyunkwan University (SKKU, Suwon-si, Gyeonggi-do, Korea), was used for the Raman measurements. X-ray photoelectron spectroscopy (XPS) analysis was carried out by ESCA2000 spectrometry (Termo Fisher Scientific, Walthan, MA, USA) using Al-Kα radiation (1468.6 eV).

## 3. Results and Discussions

The pick and place method for fabricating tBLG with a pre-defined twist angle is illustrated in Figure 1a. We used graphene synthesized on a Ge (110) wafer, which is a single crystal and monolayer [24]. After the gold pattern was fabricated on the as-grown graphene, the thermal annealing in an H_2_ atmosphere resulted in hydrogen intercalation between the graphene and the Ge substrate [26,27]. As a result, adhesion between the graphene and the underlying Ge substrates became extremely weak [24,28]. The PPC/PDMS stamp had a weak adhesion with graphene but a strong adhesion with gold, so the gold pattern and the underlying graphene could be selectively lifted (picked up) from the Ge substrate. Figure 1b shows the transferred monolayer graphene on the SiO_2_ substrate after the gold pattern was removed. Alternatively, the tBLG was produced by transferring the picked-up gold/graphene to the as-grown graphene on Ge again with a controlled twist angle. The resulting gold/tBLG on the Ge substrate was thermally annealed in a mixed H_2_ and Ar atmosphere to enhance the van der Waals interaction between the two graphene layers and weaken the adhesion between the graphene and Ge. As van der Waals interaction between graphene layers is stronger than the adhesion between graphene and Ge, the gold/tBLG was easily picked up from the Ge substrate in the same way as gold/graphene. The gold on tBLG prevents the contamination of the graphene surface that may occur during the pick and place process [29]. The transferred tBLG on the SiO_2_ substrate, after the gold pattern was removed, shows a darker color than the monolayer graphene (Figure 1c). XPS analysis was performed to check whether the tBLG was chemically contaminated during the pick and place process (Appendix A). The curve-fitted C1s spectrum of the tBLG matches well with the spectrum of high-quality graphene reported previously [30]. An Au peak in the 80~85 eV region and iodine peak in the 610~640 eV region did not appear, indicating the gold thin film was successfully removed without leaving any inorganic residues. However, in the fabrication of tBLG, graphene is evitably exposed to adsorbents such as air and hydrocarbons, which can form bubbles or blisters at the tBLG interface.

Raman analysis is a powerful tool for analyzing graphene quality and layer number [31]. In addition, when the energy of the incident photons matches the VHSs of tBLG, the intensity of the Raman G-band of graphene is significantly increased by resonance enhancement [18,32]. In Figure 2, Raman mapping images demonstrate that all of the fabricated tBLGs with interlayer twist angles of 3°, 6°, 12°, and 30° are very uniform. In the tBLG sample at 3° (Figure 2a) or 6° (Figure 2b), where the twist angle is relatively small, the intensity of the G-peak is slightly increased and the full width at half maximum (FWHM) of the 2D-peak is also increased compared to that of the monolayer graphene. In particular, when the wavelength of 532 nm is used, the G-peak intensity is at the maximum at 12° tBLG (Figure 2c). Figure 2d shows that when the twist angle becomes 30°, the intensity of the G-peak and the FWHM of the 2D peak are reduced again, which indicates that, at the tBLG with a twist angle larger than 12°, interlayer coupling between the two graphene layers becomes weak [5,33].

The SAED pattern obtained from the tBLG, designed with a 12° twist angle, demonstrates that the pick and place approach in the present work allows accurate control of the interlayer twist angle (Figure 3a). The wavelength of the Moiré pattern (*λ*) seen in the HR–TEM image of the 12° tBLG is 1.187 ± 0.005 nm, which is well-matched with the Moiré pattern wavelength of the twist angle 12° calculated using the Equation (1)
*λ*(*θ*) = *a*/[2 sin(*θ*/2)],(1)
where “*a*” is the length of graphene’s lattice constant and *θ* is the twist angle [11,34].

To further investigate the property change of the tMLG according to the number of layers, we stacked the graphene layers one by one with the same twist angles. As shown in Figure 4a, a twist quad-layer graphene with an inter-layer twist angle of 1° has a difference of 3° between the bottom layer and the top layer. However, the results of Raman spectroscopy show no difference between the 1° tBLG and the 1° tMLG. The results of the 9° tMLG also show the same with the Raman spectrum of 9° tBLG. In the case of the 30° tMLG, the bi-layer regions where the top and bottom layers meet directly without the middle layer shows a Raman spectrum similar to the AB-stacked bi-layer graphene, but the regions of the tri-layer show the same Raman spectrum as the 30° tBLG. These results show that, in the case of the same inter-layer twist angle, the Raman spectrum of the twist multi-layer graphene was not different from that of the bi-layer. We also confirmed that when three or more layers of graphene are stacked, the interaction between the top and bottom layers is subject to interference by the middle layer [12].

We fabricated the back gate field-effect transistors (GFETs) of the tBLGs to evaluate the twist-angle-dependent electrical properties of the tBLGs. Figure 5a shows typical back-gate-dependent sheet resistances for the monolayer graphene, the AB-stacked bi-layer, and the 6° and 30° tBLG. From the data, it can be observed that the sheet resistance curve of the AB-stacked bilayer graphene shows a much wider FWHM than that of the monolayer graphene, whereas the curves of the two tBLGs show FWHM similar to that of the monolayer graphene. We fabricated ten devices for each graphene type and measured the hole mobility for all of the devices (Figure 5b). The average values of the hole mobility extracted at high carrier density show that all the tBLGs (6°: 1701 cm^2^ V^−1^ s^−1^, 30°: 1561 cm^2^ V^−1^ s^−1^) have higher value than mono-layer (1298 cm^2^ V^−1^ s^−1^), while the AB-staked bi-layer (567 cm^2^ V^−1^ s^−1^) has a relatively low hole mobility. In addition, the tBLG devices with 3°, 12° and 18° angles were fabricated, and their hole mobilities were calculated as 1513, 1841, and 1814 cm^2^ V^−1^ s^−1^, respectively (Appendix A). When compared to the monolayer graphene device, the mobility was increased in all the tBLG devices, which is consistent with the results of CVD synthetic tBLG devices reported by Wu et al. [35]. In the case of AB-stacked bilayer graphene, the effective mass of the charge carriers increases, resulting in a reduced mobility compared to monolayer graphene [36]. The 3° tBLG device with a twist angle less than 5° showed relatively lower mobility than the other tBLG devices because of the interlayer coupling between two graphenes [2,7]. If the tBLG’s angle is larger than 5°, the Fermi velocity of tBLG becomes similar to that of monolayer graphene [37]. In a monolayer device, the device performance is adversely affected by charge scattering from the substrate. In the case of tBLG, however, the scattering effect in the top layer of tBLG can be screened by the underlying graphene layer, resulting in an average mobility of the top and bottom graphene layers that was higher than that of the monolayer graphene [35,38]. The decreased mobility of the 30° tBLG compared to the mobilities of tBLGs of 6°–18° twist angles may be due to the interlayer bonding by the quasi-crystal structure reported by Yao et al. [10].

## 4. Conclusions

In this study, uniform tBLG and tMLG were fabricated with a designed interlayer twist angle by the pattern pick and place method. Using the pick and place method, tBLG can be fabricated by directly stacking as-grown graphene, without the exposure to supporting materials or solution, and thereby minimizing the interlayer residues that adversely affect the properties of tBLG. We also demonstrated that, in the multilayer structure with more than two layers, the middle graphene layer interacted with the upper and lower graphene layers separately, and blocked the interaction between the upper and lower graphene. Thus, we can consider the twisted multi-layer graphene as a structure in which several tBLGs are not coupled but stacked. In addition, a tBLG structure with a large twist angle was confirmed to improve the charge carrier mobility of the graphene device. This method may enable further research on the complicated multilayer structure of 2-dimensional materials with a controlled interlayer twist angle, as well as tBLG and tMLG.

## Figures and Tables

**Figure 1 materials-12-03740-f001:**
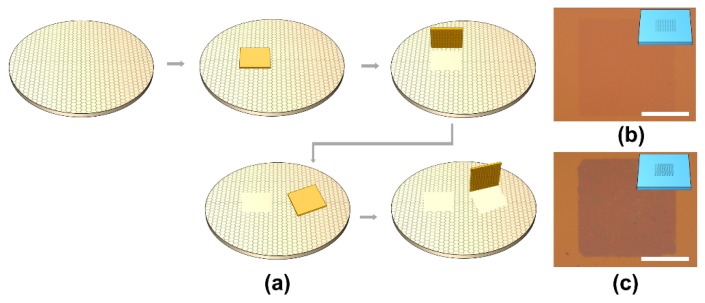
(**a**) Schematic images of the patterned transfer process. (**b**) Opticial microscope (OM) image of the pick and place monolayer graphene. (**c**) OM image of the pick and place bi-layer graphene. All scale bars are 20 μm.

**Figure 2 materials-12-03740-f002:**
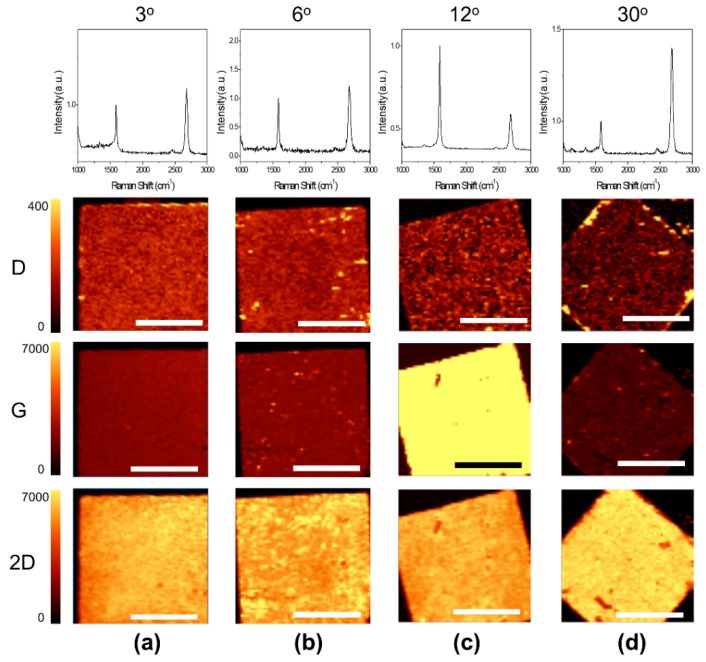
Normalized G-peak of the Raman spectrum and the intensity mapping images of the D-peak, G-peak, and 2D-peak for (**a**) 3°, (**b**) 6^o^, (**c**) 12°, and (**d**) 30° tBLG, respectively. The full width at half maximum (FWHM) of the 2D-peaks for 3°, 6°, 12°, and 30° tBLG are 39.5, 42.8, 46.5, and 38 cm^−1^, respectively. The wavelength of the Raman excitation laser is 532 nm. All scale bars are 20 μm.

**Figure 3 materials-12-03740-f003:**
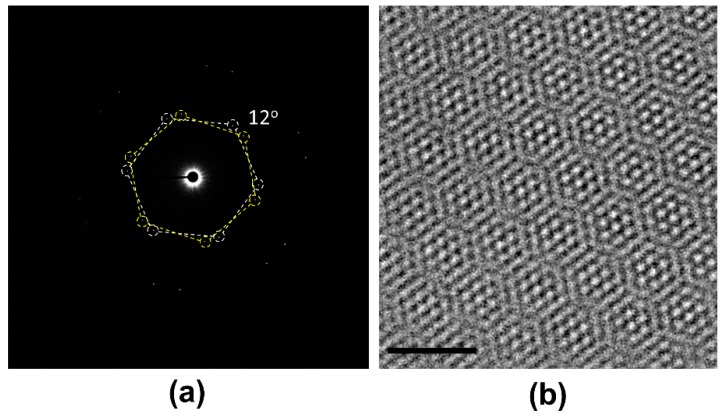
TEM analysis of the 12° tBLG. (**a**) Selected area electron diffraction (SAED) pattern showing that the two hexagons have a 12° angle difference. (**b**) HR–TEM image of the 12° tBLG’s Moiré pattern. The scale bar is 2 nm.

**Figure 4 materials-12-03740-f004:**
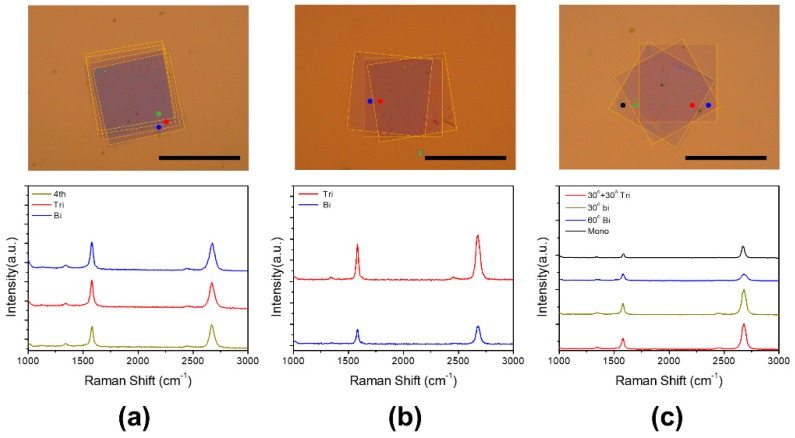
OM images and Raman spectrum of (**a**) 1° twisted quad-layer graphene, (**b**) 9° twisted tri-layer graphene, and (**c**) 30° twisted tri-layer graphene. The color points in the OM images indicate the location of the Raman data. All scale bars are 20 μm.

**Figure 5 materials-12-03740-f005:**
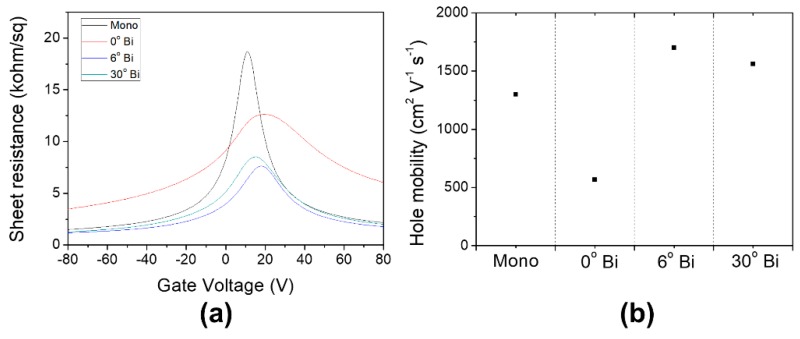
(**a**) Graph of sheet resistance vs. gate voltage, and (**b**) averages of hole mobility and sheet resistance for the mono-layer, AB-stacked bi-layer and the tBLG of 6° and 30°.

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
