# Peer review of "Pattern Pick and Place Method for Twisted Bi- and Multi-Layer Graphene"

_materials, 2019, doi:10.3390/ma12223740_

Round 1
Reviewer 1 Report
In the submitted work, authors propose and experimentally verify a smart way to stack graphene sheets one by one with a high control over the twist angle between the sheets. The proposed technique is convincing and consistent. The experimental solution and characterization of the graphene sheets assemblies are adequately performed. Moreover, the authors also report the analyses of the electrical performance of the twisted stacks of graphene sheets as a back gate of FET unit.
The overall good quality of the paper is affected by somewhat negligent attention to the text itself. The repetition of the same statement (in Section 2.3.), loose terminology (..“a” is the length of graphene lattice vector), occasional English mistakes, lack of the description of TEM specimens preparation, etc..
I do recommend the manuscript for the publication after the authors attend my comments and improve the text.
Author Response
We sincerely appreciate the effort that the reviewer 1 has taken in reviewing our manuscript. Thanks to his/her thoughtful comments, we were able to improve the quality of the manuscript. Changes have been carried out according to the comments and we hope that our revision adequately addressed the points of the comments.
Point 1: “The overall good quality of the paper is affected by somewhat negligent attention to the text itself. The repetition of the same statement (in Section 2.3.), loose terminology (..“a” is the length of graphene lattice vector), occasional English mistakes, lack of the description of TEM specimens preparation, etc.. I do recommend the manuscript for the publication after the authors attend my comments and improve the text.”
Response 1: We are grateful again for the efforts the reviewer #1 has made to review our manuscript. As the reviewer recommended, we modified the repetition of the same statement (in section 2.3, lines 90-91) and changed “lattice vector” to “lattice constant” (line 159). We have added the description of TEM specimens preparation in section 2.4 (line 91 ~ 96). We also extensively revised the manuscript to improve grammar and readability. Some of the modifications are as follows:
1) Page 2, line 90-91
The remaining polymer residue on the gold surface was removed by O2 plasma cleaning (100 sccm, 100 A, 5 min).
2) Page 2, line 90-91
2.4 Transmission electron microscopy (TEM) specimens preparation
Polymethyl methacrylate (PMMA) was spin-coated on graphene/SiO2/Si 300 nm substrate at 2000 rpm for 1 min. The spin-coated substrate was baked at 100 °C for 5 min and floated on etching solution (HF:H2O2:H2O = 1:1:1). As the SiO2 etched, PMMA/tBLG was separated from the substrate and floated on DI-water for cleaning. The PMMA/graphene membrane was scooped on a TEM grid and then PMMA was removed by using acetone vapor and solution.

Reviewer 2 Report
In the present study, authors used a method to make twisted bilayer graphene by picking up monolayer graphene from the catalyst and place it on top of another monolayer graphene with pre-designed twist angle. They also prepared more than three layers of twisted multilayer graphene to characterize the number of layers. Also, showed improved carrier mobility in tBLG compared to monolayer graphene. This method looks facile for transferring graphene from the metallic substrate and also for creating a desired twist angle, and it is interesting to a broad audience, however, there are major things needs to be addressed before considering for publication.
In the introduction, authors should make a comparison with all different transfer methods of graphene reported in the literature and explain how their method is similar/different from others. And also explain clearly why this method is novel and new. Authors also claimed that this easy pick and place method is ideal for making uniform tBLG with clean interface. To check whether the transfer process induces contamination, XPS is needed. It is important to check with XPS to understand the impurities due to metal or gas or other impurities assisted by a wet process to etch polymer and metallic supporting films. And also, maybe due to hydrocarbon gas. Randomly distributed impurities can cause inhomogeneous doping effects and thus result in hole mobilities. These effects hamper the realizations of novel phenomena and applications of graphene devices. Therefore it is important to check the impurities at each stage of transferring. Inline 118, Raman mapping images demonstrate that the fabricated tBLG of 3°, 6°, 12°, and 18° samples are very uniform but in figure 2d twist angle marked 30 degrees, not 18°. Also, the Raman G peak for 3° is asymmetric. The authors should explain the reason. Also, the background of each Raman spectra should be the same before making the comparison with peak intensity. Also, it would be nice to give the value of FWHM in each case as it is not obvious to see the difference.
Or authors could normalize with 2D peak or G peak of each spectrum to show the difference instead of showing the raw data.
What is the error bar, of measured moire wavelength of 1.2nm by TEM? Authors should also show a systematic study of sheet resistance and hall mobility for other angles too i.e. for 3°, 12°, and 18° and also explain why hall mobility is low for 30° compared to 6°.
Author Response
We sincerely appreciate the effort that the reviewer has taken in reviewing our manuscript. Thanks to the thoughtful comments of the reviewer, we were able to improve the quality of the manuscript. Changes have been carried out according to the comments and we hope that our revision adequately addressed the points of the comments.
Point 1: “In the introduction, authors should make a comparison with all different transfer methods of graphene reported in the literature and explain how their method is similar/different from others. And also explain clearly why this method is novel and new. ”
Response 1: We would like to thank the reviewer for his/her constructive comments. According to the reviewer’s suggestion, we have compared our method with the existing transfer methods of graphene. The most common method, the polymer supporting layer method, has a problem of size limitation and impurity problem caused by direct contact between graphene and polymers. The van der Waals transfer method uses a hBN instead of polymeric materials, which can reduce impurities trapped between layers. However, the van der Waals transfer allows only hBN encapsulation heterostructures and thus also limits in size due to the use of small hBN flakes. Our pattern pick and place method uses van der Waals interaction between graphene instead of polymer supporting layer or hBN flakes for fabricating tBLG. This method can reduce the impurities trap between graphene, and it is possible to fabricate twisted multilayer structures without limiting the size. We have added these contents to the introduction as follows:
1) Pages 1-2, lines 41-47
Generally, tBLG has been fabricated by folding [6, 11] or stacking [12, 13, 14] exfoliated graphene flakes. For example, a polymer supporting layer has been often used for stacking graphene flakes and thus fabricating tBLG [13]. Chen et al. reported that a single crystal graphene flake, exfoliated from highly oriented pyrolytic graphite (HOPG), was cut into two pieces with a laser, and tBLG was fabricated by stacking the two pieces with controlled twist angle [12]. These tBLG fabricated using graphene flakes has a limit in size for its practical application and impurity problem induced by direct contact with polymer [14].
2) Page 2, lines 52-56
Wang et al. used a hBN flake instead of polymeric materials as a supporting layer to transferring graphene which can reduce impurities trapped between layers [19]. Banszerus et al. reported high-performance graphene device fabricated by direct pick-up of graphene layer grown on Cu substrate using the hBN-based van der Waals transfer [20]. However, this method only allows for the hBN encapsulated heterostructures and there is a limitation in size due to the use of hBN flakes.
Point 2: “Authors also claimed that this easy pick and place method is ideal for making uniform tBLG with clean interface. To check whether the transfer process induces contamination, XPS is needed. It is important to check with XPS to understand the impurities due to metal or gas or other impurities assisted by a wet process to etch polymer and metallic supporting films. And also, maybe due to hydrocarbon gas. Randomly distributed impurities can cause inhomogeneous doping effects and thus result in hole mobilities. These effects hamper the realizations of novel phenomena and applications of graphene devices. Therefore it is important to check the impurities at each stage of transferring.”
Response 2: We performed XPS analysis of a fabricated tBLG, which confirmed that any metal or metal etching solution components used in the process do not remain in tBLG. The XPS data can be found in supplementary materials, Figure S1 and the results of the XPS analysis were described in Section 3 as follows:
1) Pages 2-3, lines 131- 136
XPS analysis was performed to check whether the tBLG was chemically contaminated during the pick and place process (Figure S1). The curve-fitted C1s spectrum of the tBLG matches well with the spectrum of high-quality graphene reported previously [30]. Au peak in the 80 ~ 85 eV region and iodine peak in the 610 ~ 640 eV region did not appear, indicating the gold thin film successfully removed without remaining any inorganic residues.
Figure S1. The XPSs results of tBLG at each C 1s, Au 4f, and I 3d
Point 3: “In line 118, Raman mapping images demonstrate that the fabricated tBLG of 3°, 6°, 12°, and 18° samples are very uniform but in figure 2d twist angle marked 30 degrees, not 18°. Also, the Raman G peak for 3° is asymmetric. The authors should explain the reason. Also, the background of each Raman spectra should be the same before making the comparison with peak intensity. Also, it would be nice to give the value of FWHM in each case as it is not obvious to see the difference. Or authors could normalize with 2D peak or G peak of each spectrum to show the difference instead of showing the raw data.”
Response 3: We corrected a typo by changing “18” to “30” in lines 143 and 148.
During the transfer process, graphene is exposed to atmospheric gases and moisture that can adsorb on the surface. It is known that the contaminants such as hydrocarbon gases and moisture can be trapped between layers in the form of bubbles or blisters which is probably the reason why the Raman G peak of 3o tBLG is asymmetric. Unfortunately, it is almost impossible to completely remove the bubbles, unless fabricating under special conditions such as a vacuum. Therefore, to address this issue, we added a sentence in line 136 ~ 137 of the revised manuscript as follows:
Page 4, lines 136-137
However, in the fabrication of tBLG, graphene is evitably exposed to adsorbents such as air and hydrocarbons, which can form bubbles or blisters at the tBLG interface.
As the reviewer recommended, we modified the Raman spectrum data in figure 2, normalized the data to G-peak intensity, and in figure 2 caption, specified that each 2D peak FWHM can be clearly compared.
Point 4: “What is the error bar, of measured moire wavelength of 1.2nm by TEM? Authors should also show a systematic study of sheet resistance and hall mobility for other angles too i.e. for 3°, 12°, and 18° and also explain why hall mobility is low for 30° compared to 6°.”
Response 4: Average of measured moire wavelength is 1.187 nm and standard error is 0.005 (in line 154).
As the reviewer suggested, we additionally fabricated the devices of tBLGs at 3o, 12o, and 18o. As we expected, tBLG at 12o and 18o showed similar mobility to 6o, but the mobility measured at 3o was similar to the value at 30o. These results are consistent with previous reports reported by Wu et al. As reported previously by Luican et al., In the case of 3o tBLG, it is considered that the Fermi velocity in the bilayer graphene is affected by the twist angle of 5o or less. In the case of 30o, we assumed that the occurrence of interlayer coupling by the quasi-crystalline structure of 30o tBLG reported by Yao et al. affects the Fermi velocity. These contents were added to lines 188-191, 194-195, and 200-201, and the back gate dependent sheet resistance graphs for 3o, 12o, and 18o tBLG devices can be found in Figure S2 of the supplementary information.
Figure S2. Back gate dependent sheet resistance graphs for 3o, 12o, and 18o tBLG devices.
Reviewer 3 Report
Manuscript title: “Pattern pick & place method for twisted bi- and 2 multi-layer graphene”
Authors: Jae-Young Lim, Hyeon-Sik Jang, Hyun-Jae Yoo, Seung-il Kim and Dongmok Whang
Recommendation: Accept
The authors are highly respected experts in their field. Their works on growth of single-crystal graphene monolayers are widely known. The manuscript reports original ideas and interesting results. I would definitely recommend to accept the manuscript for publication.
Author Response
We thank Referee #3 for his/her positive comments, strong support and favorable recommendation for our work. In order to further improve the manuscript, we added additional experimental results, such as XPS results and measurement of electrical characteristics at different twist angles. We also extensively revised the manuscript to improve grammar and readability.
Round 2
Reviewer 2 Report
The manuscript has been improved significantly after adding the suggestions and it can be accepted in the present form.